# Acceptance of Google Meet during the Spread of Coronavirus by Arab University Students

Rana Saeed Al-Maroof [1], Muhammad Turki Alshurideh [2,3], Said A. Salloum [4,5,*],
Ahmad Qasim Mohammad AlHamad [6] and Tarek Gaber [5]

1 English Language & Linguistics Department, Al Buraimi University College, Al Buraimi 512, Oman; rana@buc.edu.om
2 Department of Management, University of Sharjah, Sharjah 27272, United Arab Emirates; malshurideh@sharjah.ac.ae
3 Department of Marketing, School of Business, The University of Jordan, Amman 11942, Jordan
4 Machine Learning and NLP Research Group, University of Sharjah, Sharjah 27272, United Arab Emirates
5 School of Science, Engineering, and Environment, University of Salford, Manchester M5 4WT, UK; t.m.a.gaber@salford.ac.uk
6 Information Systems Department, University of Sharjah, Sharjah 27272, United Arab Emirates; aalhamad@sharjah.ac.ae
* Correspondence: salloum78@live.com

**Abstract:** The COVID-19 pandemic not only affected our health and medical systems but also has created large disruption of education systems at school and universities levels. According to the United Nation's report, COVID-19 has influenced more than 1.6 billion learners from all over the world (190 countries or more). To tackle this problem, universities and colleges have implemented various technologically based platforms to replace the physical classrooms during the spread of Coronavirus. The effectiveness of these technologies and their educational impact on the educational sector has been the concern of researchers during the spread of the pandemic. Consequently, the current study is an attempt to explore the effect of Google Meet acceptance among Arab students during the pandemic in Oman, United Arab Emirates, and Jordan. The perceived fear factor is integrated into a hybrid model that combines crucial factors in TAM (Technology acceptance Model) and VAM (Value-based Adoption Model). The integration embraces perceived fear factor with other important factors in TAM perceived ease of use (PEOU) and perceived usefulness (PU) on the one hand and technically influential factor of VAM, which are perceived technicality (PTE) and perceived enjoyment (PE) on the other hand. The data, collected from 475 participants (49% males and 51% females students), were analyzed using the partial least squares-structural equation modelling (PLS-SEM). The results have shown that TAM hypotheses of usefulness and easy to use have been supported. Similarly, the results have supported the hypotheses related to VAM factors of being technically useful and enjoying, which helps in reducing the atmosphere of fear that is created due to the spread of Coronavirus.

**Keywords:** Google Meet; coronavirus; perceived fear; VAM; TAM

## 1. Introduction

Coronavirus originally appeared in the market located in China and out of that place, it has been spread all over the world [1], creating the atmosphere of fear where people were strongly advised to follow certain precautions such as maintaining social distance, wearing masks, and keeping their hands clean [2–4]. Due to the importance of fear and stress factors, many studies have tackled this issue during the spread of the pandemic. One of the studies by [4] have emphasized the impact of psychological distress (anxiety, stress, and depression) within the period of the pandemic on Chinese nationals. A similar study has proven that stress, depression, and anxiety are highly evident within people,

and it significantly influences people working in the health domain [5]. Other studies have focused on making the connection between fear and stress by stating that stress is usually defined as an emotional feeling and physical tension that appears due to threat, whereas fear is closely related to anxiety and it is a kind of natural reflection to stress [6,7].

The COVID-19 pandemic not only affected our health and medical systems but also has created large disruption of education systems at school and universities levels. According to United Nation's report, COVID-19 has shown its impact on more than 1.6 billion learners from all over the world (190 countries or more) [8]. In the Arabic countries, as reported by UNICEF [9], COVID-19 has left a socio-economic impact on young people where schools and universities which fully or partially closed affect the education of more than 110 million students. Like other countries, to tackle the impact of COVID-19 on education, different colleges and universities in Arabic countries have adopted various technological tools that can help them to overcome that predicament. One of the studies has shown that Moodle was one of the effective platforms that have been used within dental schools where students have shown a positive impression [10]. A similar study by [11] has adopted a survey among university students that measures the usability of Microsoft Teams as a reference platform using the (Technology acceptance Model) TAM model. It concludes that the effectiveness of these platforms is highly evident, especially in developing countries where perceived usability is highly evident. Similarly, the effect of the perceived threat and lack of control is the main concern within the educational sector. In this respect, [12] shows that the acceptance of surveillance technologies is closely influenced by the perceived personal threat and lack of personal control. It adds that the negative feelings that people got due to the pandemic will be lessened when such technologies are used. Students' or users' satisfaction is also important within the spread of the pandemic. Users' personal factors have no influence on satisfaction but technologically available platforms help in promoting better educational atmosphere [2].

Even though previous studies have focused on technology adoption or acceptance during the spread of COVID-19 within the educational sector, it seems that there is no sufficient evidence to the impact of this technology towards teachers' and students' acceptance especially in the usage of Google Meet as a communicational and teaching platform and its effectiveness in the educational sector. Furthermore, the perceived fear factor as a dominant factor has not been investigated in a hybrid model. Based on the discussion above, it was found that the effect of perceived fear and its relation with TAM and VAM models within the educational sector has not been explored before especially within the Arab university students. Accordingly, this paper is an attempt to fill this gap and enhance teachers and educators to the more effective teaching technology during the spread of the pandemic. The reason behind adopting the hybrid model is that every model has its own constructs that will add to the acceptance of Google Meet as an online platform in an educational environment. The value-based adoption model of technology (VAM) is significantly important in examining the effect of Mobile learning technology. Mobile learning is any form of education that is meant to deal with handheld devices including smartphones and i-pads, etc. [13]. One of the interesting factors of mobile learning is its mobility, which adds the factor of flexibility in time, speed, and space [14,15]. Most Arab students are using mobile phones when they use Google Meet in every classes.

TAM model can serve the technology in its general perspective, whereas VAM can add additional perspective by considering the value of internet technology. TAM can be used to measure the effectiveness of traditional technology, whereas VAM can be used to measure users' adoption based on the value of M-Internet services and, therefore, is more effective in explaining the acceptance or adoption of M-Internet [16–18]. It is worth to mention that what sets this paper apart from previous studies during the spread of COVID-19 [2,11,19,20] is the fact that this paper adopts a hybrid model where TAM an VAM are integrated to measure the acceptance of technology in educational environment. Other papers such as [12,21,22] are implemented in non-educational environment, which

implies that it has no relation with students' perceptions or attitudes. This paper covers this where the perception and attitudes of Arab learners are the focus of this paper.

TAM model can serve the technology in its general perspective, whereas VAM can add additional perspective by considering the value of internet technology. TAM can be used to measure the effectiveness of traditional technology, whereas VAM can be used to measure users' adoption based on the value of M-Internet services and, therefore, is more effective in explaining the acceptance or adoption of M-Internet [16–18]. It is worth to mention that what sets this paper apart from previous studies during the spread of COVID-19 [2,11,19,20] is the fact that this paper adopts a hybrid model where TAM an VAM are integrated to measure the acceptance of technology in educational environment. Other papers such as [12,21,22] are implemented in non-educational environment, which implies that it has no relation with students' perceptions or attitudes. This paper covers this where the perception and attitudes of Arab learners are the focus of this paper.

The rest of this study is organized as follows. Section 2 demonstrates a comprehensive background and a summary of the literature review pertaining to the Google Meet acceptance. Section 3 tackles the theoretical framework and research model. Section 4 presents the methodology that directs the research. Section 5 discusses the study results. Section 6 provides the theoretical and practical implications and concludes the study.

## 2. Literature Review

Studies on fear and technology acceptance have been the concern of many researchers, especially after the spread of Coronavirus. Most recent studies have explored the issues of fear and technology from different perspectives and in various community sectors all over the world. All the studies have been conducted within the year 2020, either within the lockdown period or within the period when COVID-19 reaches its peak in the number of affected people. Therefore, all studies try to shed light on the possible solutions in different sectors. In terms of education at universities and colleges, it seems that studies in this sector are conducted to serve different purposes in one country, which is India. One of the studies has taken into consideration fear, anxiety, and consciousness during the lockdown period, where faculty have used different online techniques. According to this study, fear, anxiety, and consciousness have been increased during COVID-19, but the adoption of the online platform helps in decreasing the bad consequences of fear [19]. As stated by [11] shed light on the completely different aspects from an educational perspective: the perceived usability of the online learning platforms from students' perception. They reach the conclusion that the higher perception of usability leads to the adoption of the online platform during the COVID-19 period. According to this study, the ease-of-use construct within TAM has increased the level of usability among students. Other researchers have focused on a different perspective by focusing on perceived satisfaction. The study reaches the conclusion that a personal perspective has no direct influence on perceived satisfaction. What matters is the availability of the online educational platform, which has a great impact on users' perception during the pandemic in China [2]. With regard to Poland, studies have shown that fear of COVID-19 has shown a positive impact on the mode of teaching. It creates a pleasure atmosphere where students enjoy the shift from traditional classroom to virtual classrooms.

As far as the health sector is concerned, one of the studies conducted in India has explored the effect of WhatsApp in sending and receiving information within patients and physicians during the pandemic. They admitted that though there are different tools to pass the information, it seems that WhatsApp is the most common and familiar among people. This may stem from the fact that it is easy to use and can be used to transmit images and medical reports among patients and workers in the health sector. In addition to the fact that it is very safe and practical during the lockdown period [21].

Finally, within the household, researchers have explored a large scale of the population from different perspectives. One of them examines the effect of fear from COVID-19 during the closure period, where colleges, universities, institutions, and restaurants were closed

due to the spread of the pandemic. The negative feelings that appear as the result of COVID-19 create a global crisis, but the use of technology creates positive attitudes towards it and helps in reducing the ideological beliefs. Table 1 shows the main forms of fear in different sectors along with the model adopted [12]. The other perspective is held by [22] where they aim to examine the impact that track-technology on a population from 18 to 64 years old. Accordingly, they aim to study the usability of such technology in tracking people who have COVID-19 symptoms. Table 1 shows how studies are distributed among different countries such as India, China, Poland, and Vietnam from different perspectives and in different sectors.

**Table 1.** Research during the spread of pandemic and lockdown in different sectors.

| No. | Sectors | Authors | Period | Forms of Fear | Technology | Samples | Models | Country |
|---|---|---|---|---|---|---|---|---|
| 1. | Educational sector | [19] | The lockdown period | Fear, anxiety, and consciousness | Online teaching platform | College faculty | Qualitative study | India |
| | | [11] | The COVID-19 Spread period | Perceived usability of the online learning platforms | Microsoft Teams | College students | TAM | India |
| | | [23] | The spread of COVID-19 period | Perceived satisfaction of online teaching platform | Online teaching platform | College students and faculty | N/A | China |
| | | [20] | COVID-19 pandemic period | Perception of behavioral intention | Online Conferencing Tool (Zoom, Microsoft Teams, or Google Hangout) | Female college students | TAM | Vietnam |
| | | [24] | The fear of academic year loss | The psychological effect of fear and distress | E-learning | College students | N/A | Bangladesh |
| | | [25] | The pandemic outbreak | COVID-19 | N/A | Students | N/A | Poland |
| | | [26] | The spread of COVID-19 | COVID-19 | E-learning | Students | GETAMEL | Poland |
| 2. | Health sector | [21] | During the shutdown period | Fear of COVID-19 among health workers | WhatsApp as a teledermatology tool | Physicians and patients | N/A | India |
| 3. | Household sector | [12] | The closure announced by the Polish Government | Perceived threat and lack of control | Surveillance technologies | Polish people via Facebook (households) | N/A | Poland |
| | | [22] | Lockdown period | Usability of diagnostic App. | Diagnostic of COVID-19 App. | Range of population from 18 to 64 years old | UTAUT | Belgium |

Note: GETAMEL: General Extended Technology Acceptance Model for E-Learning; UTAUT: Unified theory of acceptance and use of technology.

## 3. Theoretical Model and Research Model

For the current research, the developed research model aims to integrate the perceived fear, perceived technicality, as well as the perceived enjoyment within two kinds of theoretical models, which are TAM and VAM. The proposal is that the perceived fear would influence the perceived ease of use (PEOU) and perceived technicality (PT) of Google Meet acceptance. Additionally, perceived usefulness (PU) and perceived enjoyment are expected to be influenced by the intention of using Google Meet. The proposed theoretical model is presented in Figure 1.

### 3.1. Perceived Fear

The fear factor was tackled in recent studies by [27] as an external factor that affects students' perception of technology during the pandemic. The fear from COVID-19 is

manifested in forms such as threat, anxiety, feeling of uncertainty, the risk for loved ones [19,28–32]. The feeling of fear among students has affected education and changed the platform of teaching from the physical classes to a virtual class by implementing an online platform [33]. It was a challenge to faculty and students because they rely on a different online platform such as Zoom, Skype meet up, Google classroom, etc. Instead of relying on the four-wall traditional class. In most of these institutions, faculty have shown positive attitudes towards the use of online platforms during the crisis resulting in reducing the fear effect and creating an atmosphere that resembles an actual and classical classroom [2,19].

Accordingly, this study intends to explore the effect that perceived fear (PF) has on faculty and students during the coronavirus period by focusing on other well-established constructs within TAM and VAM models. The perceived fear factor seems to be related to four constructs that can measure the acceptance of Google Meet as an online platform. The following hypotheses are made:

**H1:** *Perceived fear (PF) has a positive effect on the perceived ease of use of GM (PEOU).*

**H2:** *Perceived fear (PF) has a positive effect on the perceived technicality of GM (PT).*

### 3.2. TAM Model

TAM model has been used by many researchers recently to measure the impact of technology acceptance and adoption [34–36]. The two constructs within the TAM model, namely the perceived ease of use and perceived usefulness are crucial factors in measuring the effect of technology on its users within the educational sector [36–40]. Consequently, PEOU and PU are mediators that can be used to measure the acceptance of GM on the one hand and the effect of the PF on the other hand. Consequently, whenever users' perception of technology is marked easy to be used, it implies that users are ready to accept the technology, and they will develop positive attitudes towards technology and reduce their personal perceived fear due to the spread of Coronavirus. Similarly, the perception towards ease of use will affect the perceived usefulness (PU) and leads to technology acceptance. To apply the previous assumptions, the following hypotheses are proposed:

**H3:** *Perceived ease of use (PEOU) has a positive effect on the perceived usefulness of GM (PU).*

**H4:** *Perceived usefulness (PU) has a positive effect on the intention of using Google Meet (AGM).*

### 3.3. VAM Model

The VAM combines the theory of TAM [39] and the perceived value of [41] to measure the acceptance of technology more efficiently and effectively. The VAM model is closely related to the benefits of using technology, taking into consideration the overall judgments of the perceived value of the technology, which defines its acceptance or adoption intention. In terms of technicality, this construct measures the other technical services related to the technology, such as conference content engagement, learning, audio-visual requirements, and kinesthetic appeal [42]. From an enjoyment point of view, users who have experience with technology get immediate pleasure or joy; thus, they can consider technology as enjoyable from their own personal perspectives. They even form their own perceived enjoyment (PE) and value the technology. They will gradually develop their own attitude and accept/adopt the technology. Therefore, enjoyment can be defined as a personal feeling that is formed away from any other consequences, and it can be related to the perceived usefulness that users can develop. Having fun or enjoyment can be added as a construct to develop the acceptance theory [39,43–45]. Consequently, taking into consideration the previous assumption where perceived enjoyment is a critical factor. The current study has adopted the VAM model, which is the second part of the hybrid model; hence, the study has focused on two main constructs within VAM theory, which are perceived technicality (PT) and perceived enjoyment (PE). We therefore hypothesize:

**H5:** *Perceived technicality (PT) has a positive effect on the perceived enjoyment of GM (ENJ).*

**H6:** *Perceived enjoyment (ENJ) has a positive effect on the intention to use GM (AGM).*

The proposed research model relies on these hypotheses, as illustrated in Figure 1. The theoretical model is first given the form of a structural equation model, and then it is assessed using machine learning methods.

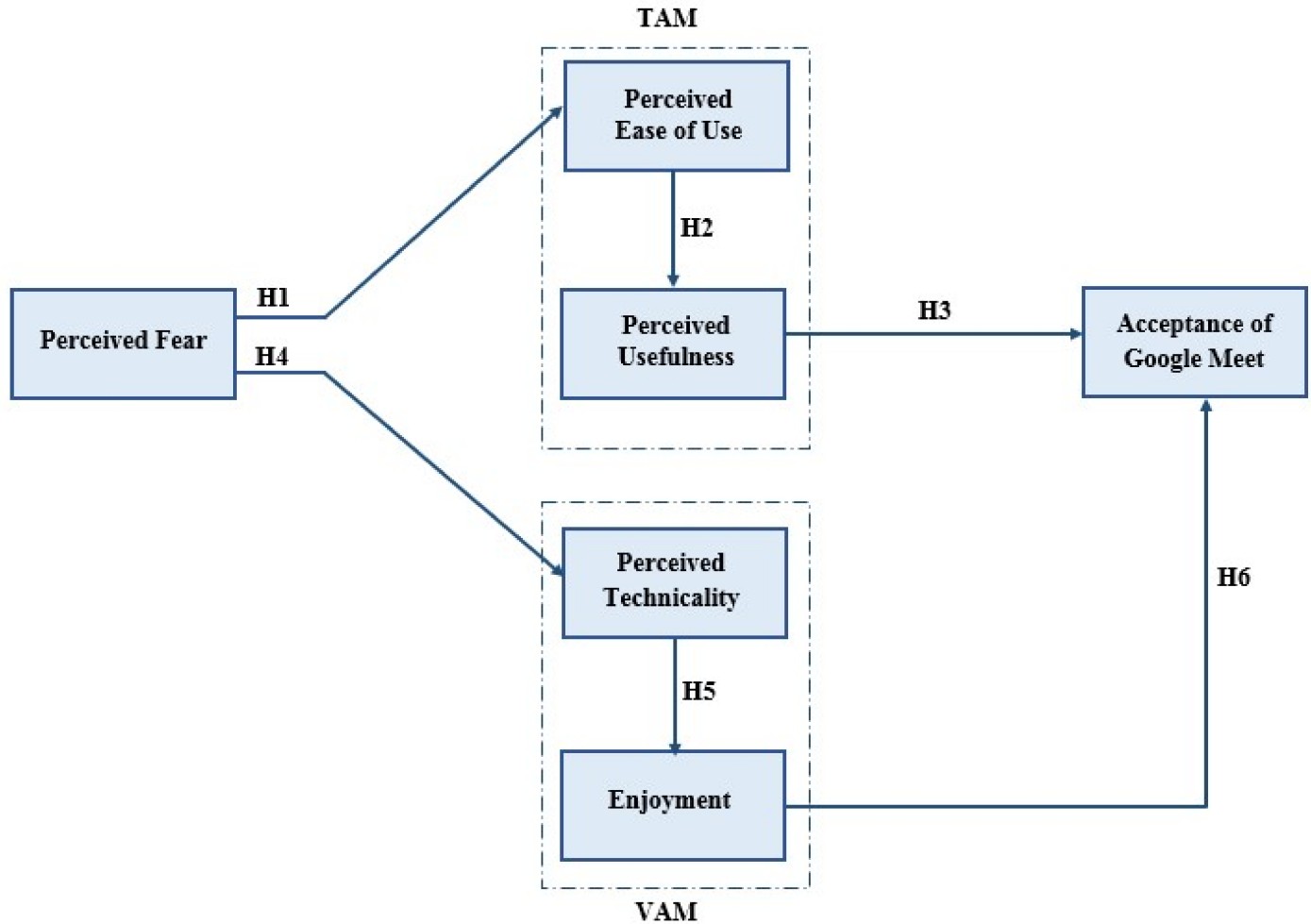

**Figure 1.** Research model.

## 4. Research Methodology

### 4.1. Data Collection

The data were collected during the fall semester 2020/2021 from mid-September to mid-October 2020, with online surveys. Five hundred questionnaires were distributed, out of which 475 respondents responded, which means that 95% was the response rate. In addition, there were 25 rejections of the questionnaires as they included some missing values. Thus, it was concluded that 475 questionnaires were helpful as they were filled correctly—the desired sample size for a population of 1500 in 306 respondents. According to [46], the responses collected were 475, and this sample size is quite large. Thus, the evaluation of the structural equation modelling is suitable for the sample size [47]. It was used to confirm the hypotheses, and the current theories played a part in establishing these hypotheses that will also incorporate the Internet of Things (IoT) context. The structural equation modelling (SEM) (Smartly Version 3.2.7, University of South Alabama, Mobile, AL, USA) was used by the group of researchers to assess the measurement model. For the better treatment path model was used.

### 4.2. Students' Personal Information/Demographic Data

The development of personal/demographic information is illustrated in Table 2. The proportion of males and females students was 49% and 51%, respectively. When talking about students' ages, 58% of the respondents were above 29 years, while the rest of 42% were the respondents who were between 18 and 29 years. The majority of the respondents were from the literate background and had university degrees. As 61% of individuals had a bachelor's degree, 24% had a master's degree, while the rest of 15% of the respondents had a doctoral degree. The "purposive sampling approach" was used when the respondents are accessible easily and are ready to volunteer, as suggested by [48]. The sample was composed of students from various backgrounds, colleges, and ages, belonging from diverse programs at different levels. Moreover, through IBM SPSS Statistics ver. 23 (IBM, New York, NY, USA), the demographic data was calculated, which was shown in Table 2.

**Table 2.** Demographic data of the respondents.

| Criterion | Factor | Frequency | Percentage |
|---|---|---|---|
| Gender | Female | 243 | 51% |
| | Male | 232 | 49% |
| Age | Between 18 and 29 | 198 | 42% |
| | Between 30 and 39 | 127 | 27% |
| | Between 40 and 49 | 78 | 16% |
| | Between 50 and 59 | 72 | 15% |
| Education qualification | Bachelor | 289 | 61% |
| | Master | 116 | 24% |
| | Doctorate | 70 | 15% |

### 4.3. Study Instrument

The study announced a survey instrument to validate the hypothesis. The survey that had an objective of measuring the six constructs in the questionnaire consisted of 24 items. Table 3 showed the sources of these constructs. The theorists modified and fixed the questions from prior researches to improve the practicality of this study. Appendix A depicts instrument development.

**Table 3.** Constructs and their sources.

| Constructs | Number of Items | Source |
|---|---|---|
| AGM | 2 | [27,39,49,50] |
| PF | 4 | [27] |
| PEOU | 4 | [39,49,50] |
| PU | 5 | [39,49,50] |
| PT | 5 | [17] |
| ENJ | 4 | [44,45] |

**Note:** AGM: acceptance of Google Meet; PF: perceived fear; PEOU: perceived ease of use; PU: perceived usefulness; PT: perceived technicality; ENJ: enjoyment.

### 4.4. A Pilot Study of the Questionnaire

The pilot study helped to find out about the reliability of the questionnaire items. This study was devised by about 50 students who were randomly chosen from a certain population. The research standards were abided with the sample size of 500 students based on 10% of the total sample size of this study. Cronbach's alpha test was used for internal reliability with the help of IBM SPSS Statistics ver. 23, to assess the evaluations of the pilot study's conclusions where the suitable results were shown for the measurement items. If we focus on the preset sequence of social science research studies, then the reliability coefficient of 0.70 is considered appropriate [51]. Table 4 shows the Cronbach alpha values for the following 7 measurement scales.

**Table 4.** Cronbach's alpha values for the pilot study (Cronbach's alpha ≥ 0.70).

| Constructs | Cronbach's Alpha |
| --- | --- |
| AGM | 0.836 |
| PF | 0.867 |
| PEOU | 0.809 |
| PU | 0.882 |
| PT | 0.865 |
| ENJ | 0.830 |

*4.5. Survey Structure*

The questionnaire survey was distributed for the research purpose. This study reached four different universities in the Oman, United Arab Emirates, and Jordan universities (*N* = 500) where they were given online surveys. These four universities are among the famous universities of Oman, United Arab Emirates, and Jordan.

The questionnaire survey that was distributed to the students [52] has three sections that were:

- The first section focuses on the personal data of the respondents.
- The second section emphasized the two items that showed common questions about Google Meet.
- The third section consisted of 22 items that showed perceived fear, perceived ease of use, perceived usefulness, perceived technicality, and enjoyment.

With the help of a five-point Likert scale, the 24 items will be assessed where these scales consist of five points: strongly disagree (1), disagree (2), neutral (3), agree (4), and strongly agreed (5).

## 5. Findings and Discussion

*5.1. Data Analysis*

To execute the data analysis in this study [53], the partial least squares-structural equation modeling (PLS-SEM) was used along with the aid of SmartPLS V.3.2.7 software (University of South Alabama, Mobile, AL, USA). To appraise the collected data [54], a dual step assessment approach, including the structural model and measurement model, was used. PLS-SEM is selected in this study due to various motives.

Primarily, the PLS-SEM is considered to be most useful when the primary research is used to develop the present study [55]. Second, with the help of PLS-SEM, the investigative studies that include complex models can be appropriately managed [56]. Third, PLS-SEM analyzes the complete model as a single unit rather than dividing it into groups [57]. Lastly, PLS-SEM offers the concurrent analysis for both measurement and structural model that will give accurate calculations [58].

*5.2. Convergent Validity*

It was suggested to consider construct reliability (including composite reliability (CR), Dijkstra-Henseler's (PA), and Cronbach's alpha (CA)) and validity (including convergent and discriminant validity) in order to appraise the measurement model [54]. Table 5 shows that Cronbach's alpha (CA) consists of the values between 0.715 and 0.833 when finding out the construct reliability. These values are bigger than the threshold value, i.e., 0.7 [51]. According Table 5, the outcomes prove that the composite reliability (CR) have values between 0.731 and 0.899; evidently, these values are bigger than 0.7, that is the recommended value [59]. With Dijkstra-Henseler's rho (PA) reliability coefficient, the theorists must assess and inform about the construct reliability. In the investigative research, the reliability coefficient ρA should indicate values of 0.70 or higher like CA and CR and the values exceed from 0.80 or 0.90 for higher levels of research [51,60,61]. As shown in Table 5, the reliability coefficient ρA of each measurement construct is above 0.70.

These findings guarantee the construct reliability, and finally, it was considered that all the constructs were accurate.

For the measurement of convergent validity, the average variance extracted (AVE) and factor loading must be verified and checked [54]. As suggested by the results of Table 5, the recommended value of 0.7 was still lower than the figures of all factor loadings. Moreover, the values ranging from 0.525 to 0.769 were given by the AVE, which were higher than the threshold value of "0.5." Based on the future results, convergent validity can be obtained effectively for all the constructs.

### 5.3. Discriminant Validity

For the measurement of discriminant validity, two approaches, the Fornell–Larcker criterion and the Heterotrait-Monotrait ratio (HTMT), were suggested to be calculated. The Fornell–Larcker condition supports the conditions as all the AVEs, and their square roots are larger than its correlation with other constructs [62] as suggested by the findings of Table 6.

Table 7 shows the HTMT ratio outcomes, which have shown that the threshold value of 0.85 remains greater than the value of each construct [63]. So, the HTMT ratio is recognized. The findings help to determine the discriminant validity. There were no problems with the assessment of the measurement model regarding its validity and reliability when dealing with the outcomes of the evaluation. Consequently, along with the in-depth use of the collected data, the structural model can be assessed.

**Table 5.** Convergent validity results which assure acceptable values (factor loading, Cronbach's alpha, composite reliability (CR), Dijkstra-Henseler's rho (PA) $\geq$ 0.70 and the average variance extracted (AVE) > 0.5).

| Constructs | Items | Factor Loading | Cronbach's Alpha | CR | PA | AVE |
|---|---|---|---|---|---|---|
| Acceptance of Google Meet | AGM1 | 0.846 | 0.715 | 0.748 | 0.744 | 0.525 |
| | AGM2 | 0.840 | | | | |
| Perceived fear | PF1 | 0.815 | 0.830 | 0.899 | 0.802 | 0.764 |
| | PF2 | 0.811 | | | | |
| | PF3 | 0.798 | | | | |
| | PF4 | 0.709 | | | | |
| Perceived ease of use | PEOU1 | 0.754 | 0.718 | 0.731 | 0.713 | 0.637 |
| | PEOU2 | 0.797 | | | | |
| | PEOU3 | 0.817 | | | | |
| | PEOU4 | 0.796 | | | | |
| Perceived usefulness | PU1 | 0.779 | 0.754 | 0.814 | 0.884 | 0.614 |
| | PU2 | 0.848 | | | | |
| | PU3 | 0.833 | | | | |
| | PU4 | 0.749 | | | | |
| | PU5 | 0.799 | | | | |
| Perceived technicality | PT1 | 0.793 | 0.726 | 0.778 | 0.719 | 0.684 |
| | PT2 | 0.869 | | | | |
| | PT3 | 0.860 | | | | |
| | PT4 | 0.889 | | | | |
| | PT5 | 0.765 | | | | |
| Enjoyment | ENJ1 | 0.868 | 0.833 | 0.895 | 0.882 | 0.769 |
| | ENJ2 | 0.715 | | | | |
| | ENJ3 | 0.836 | | | | |
| | ENJ4 | 0.869 | | | | |

**Table 6.** Fornell-Larcker scale.

|      | AGM   | PF    | PEOU  | PU    | PT    | ENJ   |
|------|-------|-------|-------|-------|-------|-------|
| AGM  | 0.903 |       |       |       |       |       |
| PF   | 0.232 | 0.847 |       |       |       |       |
| PEOU | 0.455 | 0.421 | 0.867 |       |       |       |
| PU   | 0.539 | 0.333 | 0.698 | 0.791 |       |       |
| PT   | 0.297 | 0.437 | 0.455 | 0.279 | 0.879 |       |
| ENJ  | 0.538 | 0.578 | 0.318 | 0.339 | 0.366 | 0.897 |

**Table 7.** Heterotrait-Monotrait ratio (HTMT).

|      | AGM   | PF    | PEOU  | PU    | PT    | ENJ |
|------|-------|-------|-------|-------|-------|-----|
| AGM  |       |       |       |       |       |     |
| PF   | 0.258 |       |       |       |       |     |
| PEOU | 0.468 | 0.512 |       |       |       |     |
| PU   | 0.537 | 0.681 | 0.606 |       |       |     |
| PT   | 0.261 | 0.398 | 0.344 | 0.510 |       |     |
| ENJ  | 0.378 | 0.378 | 0.312 | 0.542 | 0.442 |     |

*5.4. Model Fit*

The following fit measures are provided by the SmartPLS: In PLS-SEM, the standard root mean square residual (SRMR), exact fit criteria, d_ULS, d_G, Chi-Square, NFI, and RMS_theta show the model fit [64]. The difference between the experienced correlations and model implied correlation matrix [65] by the SRMR, where the values that are lesser than 0.8 are recognized as a good model fit measures [56], and the NFI values greater than 0.90 show a good model fit [66]. The NFI is a ratio of the Chi2 value of the proposed model to the null model or benchmark model [67]. The larger the parameters, the superior the NFI, and therefore, NPI is not suggested as a model fit indicator [65]. The squared Euclidean distance, d_ULS, and the geodesic distance d_G are two metrics that offer discrepancy connecting empirical covariance matrix and covariance matrix implied by composite factor model [65,68]. only for the reflective models, the RMS theta is valid and estimates the degree of outer model residuals correlation [67]. The nearer the RMS theta value is to zero, the better the PLS-SEM model, and their values less than 0.12 are recognized as a good fit, while any other value represents a lack of fit [69]. According to [70], the saturated model judge link between all constructs and the estimated model takes total effects and model structure into consideration.

According to Table 8, the RMS_theta value was 0.077, which means that the concerned goodness-of-fit for the PLS-SEM model was large enough to display global PLS model validity.

**Table 8.** Model fit indicators.

|            | Complete Model | | |
|------------|-----------------|---|-----------------|
|            | Saturated Model | | Estimated Model |
| SRMR       | 0.037           | | 0.034           |
| d_ULS      | 0.764           | | 1.328           |
| d_G        | 0.517           | | 0.546           |
| Chi-Square | 474.747         | | 473.479         |
| NFI        | 0.840           | | 0.838           |
| Rms Theta  | | 0.077 | |

*5.5. Hypotheses Testing Using PLS-SEM*

Along with the Smart PLS having maximum likelihood estimation, the structural equation model was used to measure the interdependence of the structural model's various

theoretical constructs [56,71,72]. Thus, the proposed hypotheses were evaluated. The model had high predictive power, as illustrated in Table 9 and Figure 2. The $R^2$ values for acceptance of Google Meet, perceived ease of use, perceived usefulness, perceived technicality, and enjoyment were found to be above 0.67; hence, these constructs' predictive power is considered high [73]. For each of the established hypotheses depending on the generated results through the PLS-SEM technique, Table 10 provides the beta ($\beta$) values, *t*-values, and *p*-values. Most of the theorists have supported all hypotheses. According to the data analysis, hypotheses H1, H2, H3, H4, H5, and H6 were supported by the empirical data.

The first hypothesis shows the relationship between perceived fear (PF) and perceived ease of use (PEOU) ($\beta$ = 0.536, t = 2.346). The result of this hypothesis reveals that perceived fear has a significant positive impact on the perceived ease of use of Google Meet. Thus, H1 is supported. The second hypothesis describes the correlation between perceived fear (PF) and perceived usefulness (PU) ($\beta$ = 0.659, t = 3.527). This hypothesis indicates that perceived fear has a significant positive effect on the perceived usefulness of using Google Meet. Therefore, H2 is supported.

The third hypothesis indicates the relationship between perceived ease of use (PEOU) and perceived usefulness (PU) ($\beta$ = 0.783, t = 15.858). This hypothesis suggests that perceived ease of use has a significant positive influence on the perceived usefulness of using Google Meet. Hence, H3 is supported. The fourth hypothesis characterizes the correlation between perceived usefulness (PU) and the intention of using Google Meet (AGM) ($\beta$ = 0.458, t = 19.577). This hypothesis demonstrates that perceived usefulness has a significant positive influence on the intention of using Google Meet (AGM). Thus, H4 is supported. The fifth hypothesis suggests the relationship between perceived technicality (PT) and perceived enjoyment (ENJ) of using Google Meet ($\beta$ = 0.378, t = 18.330). This hypothesis exhibits that perceived technicality has a significant positive influence on perceived enjoyment of using Google Meet. Therefore, H5 is supported. The sixth hypothesis reveals the correlation between perceived enjoyment (ENJ) and the intention of using Google Meet (AGM) ($\beta$ = 0.484, t = 16.239). This hypothesis describes that perceived enjoyment has a significant positive effect on the intention of using Google Meet. Hence, H6 is supported.

**Table 9.** $R^2$ of the endogenous latent variables.

| Constructs | $R^2$ | Results |
|:---:|:---:|:---:|
| AGM | 0.698 | High |
| PEOU | 0.757 | High |
| PU | 0.743 | High |
| PT | 0.702 | High |
| ENJ | 0.739 | High |

**Table 10.** Hypotheses-testing of the research model (significant at ** $p \leq 0.01$, * $p < 0.05$).

| H | Relationship | Path | *t*-Value | *p*-Value | Direction | Decision |
|:---:|:---:|:---:|:---:|:---:|:---:|:---:|
| H1 | PF -> PEOU | 0.536 | 2.346 | 0.043 | Positive | Supported * |
| H2 | PEOU -> PU | 0.659 | 3.527 | 0.029 | Positive | Supported * |
| H3 | PU -> AGM | 0.783 | 15.858 | 0.001 | Positive | Supported ** |
| H4 | PF -> PT | 0.458 | 19.577 | 0.000 | Positive | Supported ** |
| H5 | PT -> ENJ | 0.378 | 18.330 | 0.000 | Positive | Supported ** |
| H6 | ENJ -> AGM | 0.484 | 16.239 | 0.002 | Positive | Supported ** |

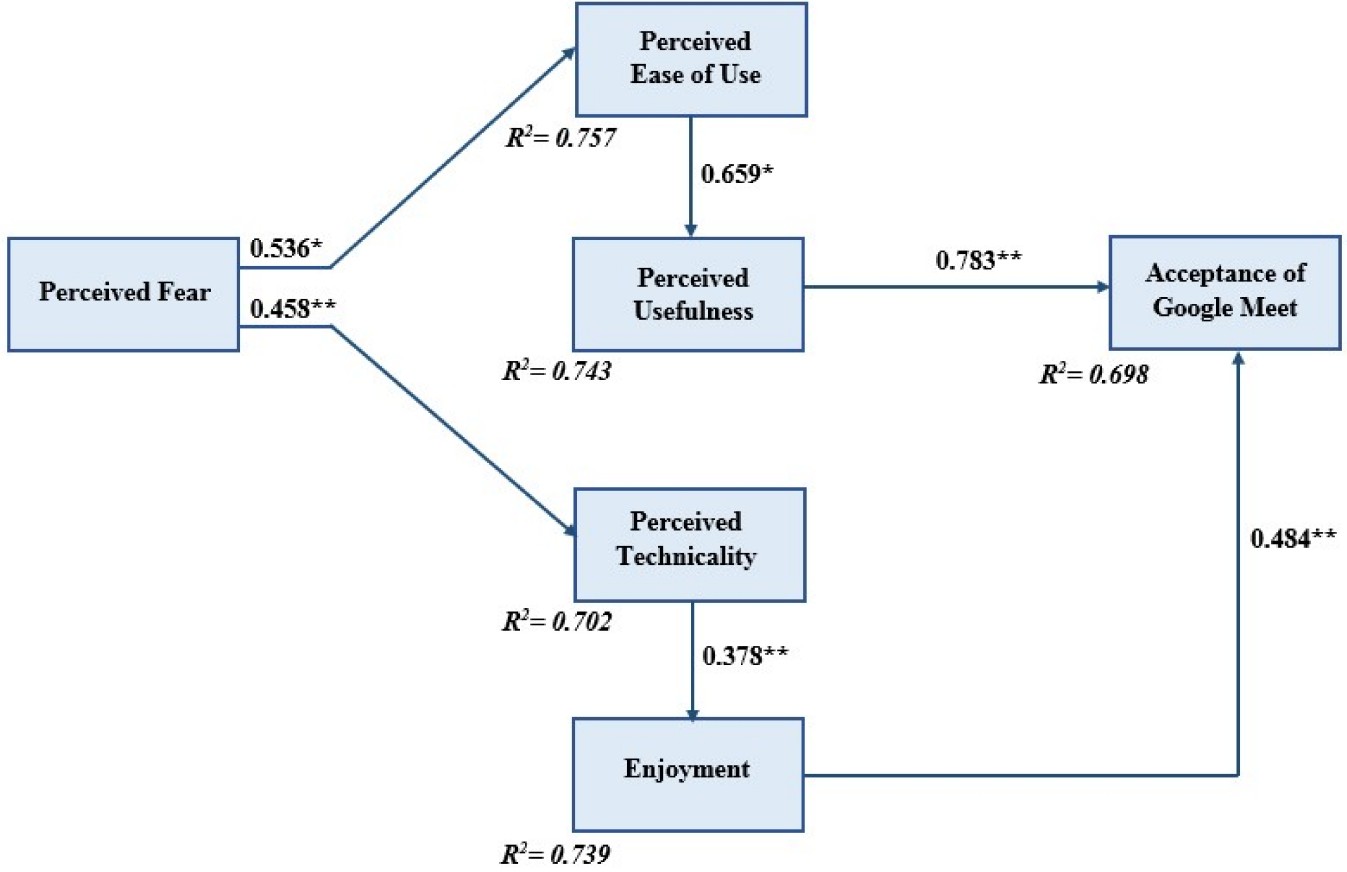

**Figure 2.** Path coefficient of the model (significant at ** $p \leq 0.01$, * $p < 0.05$).

## 6. Conclusions and Future Work

This empirical study affirmed that students and teachers have suffered from stress and fear during Coronavirus's spread. This study offers encouraging alternatives to reduce the effect of fear and enhance students learning process. Simultaneously, the literature review has provided evidence with the effect of the online platform during the pandemic asserting that the availability of a suitable online platform can help overcome difficulties during the spread of Coronavirus [24]. In this study, the hybrid model would provide a deeper understanding of certain technological factors in facilitating the process of learning and the development of college students' educational skills during the COVID-19 outbreak. In other words, the findings seem to be in line with other previous studies by [2,11], who confirms that online platforms could escalate and enhance both the perceived satisfaction and perceived usability by students of online platforms. Similarly, a study by [19] has emphasized the effect of online learning platforms on reducing both fear and anxiety that were highly evident among students during the pandemic spread.

Concerning the perceived fear factor, studies by [2,11,19,20] tackled the fear factor from different perspectives in the educational sector. According to these studies, fear has different forms, such as fear of stress, fear of loss of the academic year, fear of usefulness, fear of satisfaction, fear of usability, etc. All these fears can be reduced when the students are using highly useful technology, convenient and enjoyable. The result seems to agree with a previous study by [27], which has shown that the perceived fear is reduced because of the high degree of PU and PEOU, creating an encouraging environment to pursue studying and attend classes regularly.

Within the TAM model, the research results are in agreement with the literature concerning the effectiveness of TAM variables [38,74,75]. It has been shown that students' intention to accept technology during the pandemic is higher when the online platform provides good facilities to improve the educational process. Likewise, Google as an

online platform, has been highly evaluated as useful due to the fact that it is easy to use. Similarly, studies by [20,27] are consistent with the results that are found in the current study regarding PU and PEOU since PU and PEOU have positively affected the acceptance of Google Meet and helped in forming a conceptual framework to the students' reaction of fear during the spread of Coronavirus.

Within the VAM model, the VAM model deals with factors that influence value perception and how to value perception affect technology acceptance [17]. This fact adds a new conceptual dimension that helps us to have a clear understanding of students' perception of technology. The study results are in line with previous research concerning the factors of perceived technicality and enjoyment. It has been proven that acceptance of technology has a close relation with the perceived value to its benefits especially the technical usefulness and enjoyment. A study by [42] has shown that the usefulness of the technical aspect and enjoyment can enhance the adoption process. Lau's et al.'s results have reinforced and supported in this study as it has been approved that technically useful online platform of learning affects positively the acceptance of Google Meet as an online platform. Similarly, the enjoyment factor urges students to use the technology repeatedly resulting in accepting the technology.

Future studies will tackle different aspects that have not been covered in this paper. One of the aspects is to include a classmate (Social Norm) effect to reduce the effect of perceived fear within a hybrid model. Future studies may also include a reference to other sectors such as economic (banking) and health sectors within a hybrid model that shows both actual and perceived value of the online platform. The other aspect that the future can focus on is the inclusion of a different online platform user sample. Furthermore, variations in samples by creating a comparison among male and female users and its effect on their fear can be useful to show how gender difference is affected by the fear factor. Finally, the current study has included a sample of students and teachers, whereas other studies can include physicians, doctors, nurses, customers, etc.

### 6.1. Practical Implications

As far as practical consequences are concerned, this study is the first one to report on the effect of fear on the teaching and learning process among students in Arab countries during the spread of the pandemic by making a connection between fear factor on the one hand and TAM and VAM on the other hand. This type of study will help educational institutions recognize the effectiveness of online platform in delivering the classes, communicating with students, establishing strong relations among teachers and students who never met due to the pandemic or meeting each other for a short time. The ability to maintain good relation among teachers and students relying on the online platform is a sensitive and critical issue. Accordingly, this study may provide a solution to build up a mutual relation between teachers and their students. Once this approach of learning has proven to be effective, colleges and universities will think seriously of implementing the same learning approach even after the pandemic.

The study's findings would focus on the understanding of knowledge associated with other effective constructs within this teaching approach, including ease of use, usefulness, perceived value, and enjoyment. In other words, the conceptual model that is developed in this study can measure the effectiveness of usability, easiness, value, and enjoyment, which have a positive role in the reduction in fear urging educational institutions to implement online platforms due to its unique feature and its role in reducing the fear. They could finally achieve their educational goals. To put more specifically, the perceived factor is reduced whenever the students and teachers implement an easy-to-use online platform that is technically and personally useful. Furthermore, the enjoyment factor affects the communication process among teachers and students, resulting in better learning styles, efficient e-course material, and positive perception on behalf of students and ultimately improves the teaching and learning process.

*6.2. Limitations of the Study*

This study is limited in scope, selected sample, and application. The scope is limited to the perceived factor and its relation to the hybrid model of TAM and VAM. The effect of social norm and the precedence of a classmate was not part of the scope of the current study. Besides, the survey was distributed among Arab learners (Oman, United Arab Emirates, and Jordan) who used Google Meet during the pandemic but in selected Arab countries. Google Meet as an online platform has been widely used in the educational atmosphere all over the world, but this study is limited to Arab countries where students have the Arabic language as their native language. Finally, the study is limited in its application within the educational sector. The survey was distributed among students at colleges and universities. The study has not tackled the differences among gender in its application because we based our results away from gender difference.

**Author Contributions:** Formal analysis, S.A.S.; project administration, S.A.S.; resources, M.T.A.; software, M.T.A. and S.A.S.; supervision, A.Q.M.A.; writing—original draft, R.S.A.-M.; writing—review and editing, T.G. All authors have read and agreed to the published version of the manuscript.

**Funding:** This research received no external funding.

**Institutional Review Board Statement:** Not applicable.

**Informed Consent Statement:** Not applicable.

**Data Availability Statement:** The data presented in this study are available on request from the corresponding author.

**Conflicts of Interest:** The authors declare no conflict of interest.

## Appendix A

*Appendix A.1. Instrument Development*

Appendix A.1.1. Google Meet Acceptance (AGM)

- AGM1: Using Google Meet is highly recommended in my study during the spread of Coronavirus.
- AGM2: Using Google Meet with my teachers and classmates develops my learning abilities during the spread of Coronavirus.

Appendix A.1.2. Perceived Fear (PF)

- PF1: Using Google Meet increases my fear during the spread of Coronavirus.
- PF2: Using Google Meet is enjoyable therefore it reduces my fear.
- PF3: Using Google Meet is technically easy therefore it provides a chance to learn during the lockdown period.
- PF4: Using Google Meet is easy and useful therefore it gives me the chance to communicate with my teachers and classmates during the spread of Coronavirus.

Appendix A.1.3. Perceived Ease of Use (PEOU)

- PEOU1: Using Google Meet is easy.
- PEOU2: Using Google Meet makes communication with my teacher easy.
- PEOU3: Using Google Meet makes my interaction with my classmates more effective and easy.
- PEOU4: Using Google Meet makes it easy to do what I want to do in my study.

Appendix A.1.4. Perceived Usefulness (PU)

- PU1: Using Google Meet enables me to complete my homework more quickly
- PU2: Using Google Meet is useful to conduct my quizzes and exercises.
- PU3: Using Google Meet enhances the quality of my study.
- PU4: Using Google Meet enables me to understand new information easily.

- PU5: Overall, Google Meet is useful in my study during the spread of Coronavirus.

Appendix A.1.5. Perceived Technicality (PT)

- PT1: Using Google Meet during my daily classes is technically easy.
- PT2: Using Google Meet enables me to be connected with my classmate immediately without any technical problems.
- PT3: Using Google in my daily class is complicated and takes along of time.
- PT4: Using Google Meet in daily has may audio and visual problems.
- PT5: Overall, using Google Meet is very simple and attainable in my daily classes.

Appendix A.1.6. Perceived Enjoyment (ENJ)

- ENJ1: Using Google Meet is fun.
- ENJ2: Using Google Meet is pleasurable.
- ENJ3: Using Google Meet gives me a lot of enjoyment.
- ENJ4: Using Google Meet makes me excited.

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
