# Peer review of "Acceptance of Google Meet during the Spread of Coronavirus by Arab University Students"

_informatics, doi:10.3390/informatics8020024_

Round 1

Reviewer 1 Report

The paper explores the effect of Google Meet acceptance among Arab students during the pandemic in Oman, UAE and Jordan.

The topic of the paper is interesting, but its structure leaves open questions about the scientific contribution of research. I believe that the article could pass as a preliminary statement, or pre-research as a basis for some future research, but not at all like the original scientific article. The research is a pretty „light“ and I think it should be further upgraded and/or better explained its key parts for acceptance in this journal.

The introductory part of the paper should provide a review of the literature as well as a review of scientific papers that you consider relevant papers for the development of your models. In all the key sections, you re-list the relevant papers, which is quite unusual given that the methodology and results should be your original scientific contribution.

  1. Please clearly separate in the paper the part where you present what has already been done, from what you have created. The chapter where you present your model is not written in accordance with the scientific paper. It is necessary to clearly point out your scientific contribution, which undoubtedly shows your original contribution. 
  2. The chapter "Methods and methodology" is insufficiently well explained in terms of describing the methodology. Given that the survey is a key part of the whole research, it is necessary to clearly state how the questionnaire is structured. It is not only the choice of the question that is considered but also the methodology used in defining it. The structuring of the survey, and later the analysis, is a pretty complex process and much depends on the way the survey is defined. Do the authors have previous experience in defining the surveys or did they hire an expert? Was the survey tested at the very beginning on a small number of respondents who indicated whether everything was clear and whether it was statistically (metrically) tested? The theoretical parts of the survey should be presented in the chapter "Methods and methodology". Equally, the results need to be presented in the results section.

Technical disadvantages: 

  1. Why does the line count in an article always start from number 1 for each new chapter?
  2. It is necessary in the abstract and introduction, write the full name of the TAM and VAM models the first time when they appear in the text.
  3. There is two figures two in the paper.
  4. The quality of the figures is unacceptable for the publication.

Author Response

The authors are really very grateful to the feedback and comments raised by the reviewer which really assist them to significantly enhance this work and its presentation. The productive and valuable remarks enable us to update many parts of the paper as shown by the responses to each comment. Our responses are mentioned below under each comment raised by the reviewer and it is written in (Times New Roman, red color). Besides, all the updated parts in the manuscript were highlighted in yellow color in order to be easily tracked by the reviewers.

Reviewer 2 Report

Thank you very much for the thorough research you have conducted on this very urgent topic. For me, the manuscript is organized in a very clear manner, it is well-structured. The methodology is clear.

My comments are on minor issues and are given below.

1. I am not sure I understand what "M-Internet" is. Is it Mobile-Internet? I suggest you explain it once since such abbreviation is not commonly used.

2. You have mentioned that studies on universities are mostly focused on India. But, for instance, in Poland, a lot of research was conducted in the area of COVID-19 and its influence on higher education. Research papers like "Students’ Acceptance of the COVID-19 Impact on Shifting Higher Education to Distance Learning in Poland" and "COVID-19 and Higher Education: First-Year Students’ Expectations toward Distance Learning" also apply the TAM model for the research.

3. Language: I highly recommend reviewing the text from the point of view of punctuation. There are quite a few commas missing, which in some cases is confusing when reading a sentence. 
And, please, consider a slight review of English: there are a few words in the text that may be repeated even in one sentence. Try searching for synonyms.
Also, consider the option "the atmosphere of fear" instead of "fear atmosphere". The first one sounds more common and readable.
One more thing here - it is always "The United Nations".

Author Response

(The authors gave the same response as above.)

Reviewer 3 Report

You present a good paper in which you raise the question of the technological acceptance of Google Meet in a pandemic situation.

You pose an interesting, necessary and relevant research problem.

Your work is well structured. You present an introduction, a theoretical review, a set of hypotheses, a research model and then analyse the data with PLS-SEM.

I would like to make some formal suggestions in relation to your work, which I have already said that I liked and found good.

I would ask that the first time the models TAM (Technology Acceptance Model) and VAM (Value-based Adoption Model) appear, you indicate which models they are, to make it easier for the uninitiated reader to understand them, and that you indicate the authors who developed them, on the one hand Davis (1989) and on the other hand Kim, Chan, and Gupta (2007).

You are not consistent in the way you reference the authors both in the text and in the bibliography.

For example you reference: (R. A. S. Al-Maroof & Al-Emran, 2018; Fred D Davis, Bagozzi, & Warshaw, 1989; Li & Yu, 2020; Scherer, Siddiq, & Tondeur, 2019). Remove initials or names in bibliographic citations. Follow the journal's guidelines for authors.

Presentan dos citas para un mismo trabajo en la bibliografía.

Davis, F. D. (1989). Perceived Usefulness, Perceived Ease of Use, and User Acceptance of Information Technology. MIS Quarterly, 25

13(3), 319-340. https://doi.org/10.2307/249008 26

Davis, Fred D. (1989). Perceived usefulness, perceived ease of use, and user acceptance of information technology. MIS Quarterly, 27

319-340.

I recommend that you be more careful about the formal aspects. Please check your bibliography, be consistent with the initial names... I recommend that you use a citation management program.

In the introduction you state: "The value-based adoption model of technology (VAM) is significantly important in examining the effect of M-Internet based technology."

My question is why do you highlight M-Internet, Internet Technology. I understand that it can be used for any technology.

You continue "TAM model can serve the technology in its general perspective, whereas VAM can add additional perspective by considering the value of internet technology. TAM can be used to measure the effectiveness of traditional technology whereas VAM can be used to measure users' adoption based on the value of M-Internet services and therefore is more effective in explaining the acceptance or adoption of M-Internet (Kahneman & Tversky, 2013; Kim, Chan, & Gupta, 2007; Neumann & Morgenstern, 1947)".

Why do they devote this paragraph to justifying M-Internet when the dude Google Meet service is not necessarily M-Internet?

I would ask you to try to theoretically strengthen the hypotheses you propose. In relation to the fear of COVID 19, I propose the following papers:

Nguyen, H. T., Do, B. N., Pham, K. M., Kim, G. B., Dam, H. T., Nguyen, T. T., ... & Duong, T. V. (2020). Fear of COVID-19 scale—associations of its scores with health literacy and health-related behaviors among medical students. International Journal of Environmental Research and Public Health17(11), 4164.

Velicia-Martin, F., Cabrera-Sanchez, J. P., Gil-Cordero, E., & Palos-Sanchez, P. R. (2021). Researching COVID-19 tracing app acceptance: incorporating theory from the technological acceptance model. PeerJ Computer Science7, e316.

Vanni, G., Materazzo, M., Pellicciaro, M., Ingallinella, S., Rho, M., Santori, F., ... & Buonomo, O. C. (2020). Breast cancer and COVID-19: the effect of fear on patients' decision-making process. in vivo34(3 suppl), 1651-1659.

What I am proposing minor revisions for you to change.

Author Response

(The authors gave the same response as above.)

Round 2

Reviewer 1 Report

The manuscript is improved. For this reason, I recommend to accept it as it is for the next step.